# Radiographic Biomarkers for Knee Osteoarthritis: A Narrative Review

**DOI:** 10.3390/life13010237

**Published:** 2023-01-14

**Authors:** Ahmad Almhdie-Imjabbar, Hechmi Toumi, Eric Lespessailles

**Affiliations:** 1Translational Medicine Research Platform, PRIMMO, University Hospital Centre of Orleans, 45100 Orleans, France; 2Department of Rheumatology, University Hospital Centre of Orleans, 45100 Orleans, France

**Keywords:** knee osteoarthritis, biomarker, radiography, prediction, incidence, progression, total knee replacement

## Abstract

Conventional radiography remains the most widely available imaging modality in clinical practice in knee osteoarthritis. Recent research has been carried out to develop novel radiographic biomarkers to establish the diagnosis and to monitor the progression of the disease. The growing number of publications on this topic over time highlights the necessity of a renewed review. Herein, we propose a narrative review of a selection of original full-text articles describing human studies on radiographic imaging biomarkers used for the prediction of knee osteoarthritis-related outcomes. To achieve this, a PubMed database search was used. A total of 24 studies were obtained and then classified based on three outcomes: (1) prediction of radiographic knee osteoarthritis incidence, (2) knee osteoarthritis progression and (3) knee arthroplasty risk. Results showed that numerous studies have reported the relevance of joint space narrowing score, Kellgren–Lawrence score and trabecular bone texture features as potential bioimaging markers in the prediction of the three outcomes. Performance results of reviewed prediction models were presented in terms of the area under the receiver operating characteristic curves. However, fair and valid comparisons of the models’ performance were not possible due to the lack of a unique definition of each of the three outcomes.

## 1. Introduction

Although there is a great body of evidence that research has progressed in developing novel therapeutic approaches [1], there are still no efficacious disease modifying treatments for osteoarthritis (DMOADs) [2]. Joint space narrowing (JSN) or measurement of loss of tibiofemoral joint space width (JSW), in spite of efforts to improve radiographic techniques, are considered to have several methodological shortcomings [3,4], such as poor sensitivity to change [5]. However, radiographic JSN remains the measure of choice for DMOAD registration studies [6]. The current gold standard for diagnosing OA is the plain radiograph and it is possible to make a clinical diagnosis of knee OA (KOA) in the absence of definite radiographic disease as recommended by EULAR; imaging is not required to make the diagnosis in patients with typical presentation of OA [7].

For KOA, a variety of moderate to strong risk factors have been reported to influence the incidence of KOA, including age, gender, obesity and being overweight [8,9,10]. However, as concluded in a systematic review on evidence on risk factors for KOA in older adults, obesity has a slightly larger effect on incidence of KOA pain than being overweight [11].

Other factors play a role in the development of KOA including, but not limited to, muscle weakness, previous knee trauma and Knee malalignment [8,12]. Including such clinical parameters in KOA risk assessment models has produced only moderate success [13]. However, adding imaging or biological biomarkers provided better results [14,15,16].

Recent research has focused mainly on magnetic resonance imaging (MRI) as this imaging technique can directly evaluate not only cartilage tissue, albeit sometimes imperfectly [17], but also the intra and periarticular tissue involved in the global failure of the joint [18,19,20]. However, operator expertise and scanner time result in a high cost, limiting its availability for routine management in KOA. Plain radiography is readily accessible, easily acquired and relatively inexpensive. Thus, keeping in mind the need to improve the quantitative and qualitative assessment of JSN, research has been carried out to develop in parallel novel imaging biomarkers of interest in the field of KOA.

Beyond cartilage and synovial tissue, the subchondral bone tissue is recognized to play a central role in the pathophysiology of KOA [20,21].

Much research interest has focused on cartilage tissue imaging but reliable imaging biomarkers of other tissues involved in the global failure of the entire joint such as the infrapatellar fat pad [22,23] or the subchondral trabecular bone could be relevant to help in understanding the pathophysiology of KOA [24,25].

Trabecular bone texture (TBT) analysis of subchondral bone on knee plain radiographs has been described as a potential new and relevant imaging biomarker in the field of osteoarthritis [26,27]. There is a growing body of literature showing its usefulness in the early detection of radiograph KOA, the prediction of KOA progression and the prediction of KOA course until total knee arthroplasty (TKA).

A challenge for researchers is to identify biomarkers among OA patients that can predict progressors and non-progressors long in advance. In the cohorts selected in this review, for example, only 13–48% of the OA patients experienced radiographic progression within 4 years [28,29].

This narrative review aimed to summarize publications with original data on radiographic imaging markers to permit early diagnosis, prediction of KOA progression and prediction of TKA risk.

The improvement in medical imaging technologies has helped the scientific osteoarthritis community to potentially provide clinicians with prognostic data from conventional knee X-ray datasets. Due to the role of the subchondral bone and its remodeling status in KOA progression, texture analysis of the tibial subchondral bone and knee joint severity assessments have been widely studied for their association with KOA progression.

## 2. Methods

This review highlights original research articles published on the radiographic biomarkers used for the prediction of knee osteoarthritis-related disease.

Literature searches were conducted in the PubMed electronic database for research publications describing human studies on radiographic imaging biomarkers used for the prediction of knee osteoarthritis-related disease, in the period between 1 January 1991 and 31 October 2022. The search keywords and Medical Subject Headings are detailed in Appendix A.

The PubMed database search was limited to English language full-text. Exclusion criteria included studies related to other than human knee osteoarthritis imaging techniques without radiography, and studies without prognosis/prediction/risk assessment for KOA. Exclusion criteria also included non-peer-reviewed articles, dissertations, abstracts, conference proceedings, commentaries, reviews and letters to the editor.

The selection process was performed independently by two authors (A.A.-I., E.L.). Agreement was achieved by discussion.

In this review, the following elements were obtained from each study:Name of the study;Number of subjects (and images) included;Definition of radiographic progression;Inclusion criteria;Radiographic biomarkers investigated;Performance results (if possible, the area under the receiver operating characteristic (ROC) curve (AUC) score, because AUC is usually used to indicate the overall accuracy of a test according to its sensitivity and its specificity).

## 3. Results

The PubMed database search resulted in 545 papers. After scanning the titles and abstracts in Zotero, 32 studies were found eligible for the comprehensive review. The reference lists of these studies were further screened to identify any relevant missing studies. Consequently, an additional six studies were included in the present review. After reading the full text, 14 of the studies were found not to meet the eligibility criteria, see Figure 1, and hence were excluded. The lower number of studies on TKA risk prediction can be explained by the complexity of defining TKA as an endpoint due to the subjectivity of doctors’ and patients’ opinions in the final decision of TKA [30,31,32]. None of the resulting studies used radiographic markers based on quantitative computerized tomography imaging.

Two authors (A.A.-I., E.L.) independently classified the selected studies. Any disagreements between the reviewers were resolved by a consensus meeting with the third author (H.T.). The three criteria used in the classification process were:Prediction of radiographic KOA incidence;Prediction of KOA progression;Prediction of total knee arthroplasty risk.

As illustrated in Table 1, the number of studies included in the prediction of KOA incidence, KOA progression and TKA risk was 10, 12 and 5 studies, respectively. It should be noted that three studies included results on both KOA incidence and progression [33,34,35].

### 3.1. Prediction of KOA Incidence

Several studies have investigated the ability to predict the onset (incidence) of knee OA based on observation of the subchondral bone of the tibial plateau. The main specifications of the 10 studies detected for this category are summarized in Table 2. Among these 10 studies, three were based on TBT analysis [33,37,39]. TBT analysis has been identified as an imaging biomarker that provides information on trabecular bone changes due to Knee Osteoarthritis (KOA) [26,29]. The other seven studies were based mainly on geometrical parameters [14,35,36,40], on the presence of patellofemoral joint osteoarthritis (PFJOA) [34], on JSW measurement [38] and on KL grade [15].

The studies in [33,34,37,39] included patients with a Kellgren–Lawrence (KL) grade = 0 at baseline, whereas the studies in [25,35,36] included, in addition, patients with KL = 1.

The definition of the KOA incidence was based either on the increase in JSN or KL, or on the decrease in JSW.

#### 3.1.1. Increase in Medial Joint Space Narrowing

The performance of baseline TBT-based biomarkers was examined for predicting incident radiographic KOA in terms of the increase in OARSI medial JSN (mJSN) score (∆mJSN > 1) [14,25,33,37].

In a set of 203 X-ray knee images (from 105 subjects from Lund University study), a method was proposed for the analysis of TBT using the Signature Dissimilarity Measure (SDM) [33] generating a set of three descriptors: roughness, degree of anisotropy and direction of anisotropy. In this 4-year follow-up study, results illustrated the ability of the SDM-based model to predict incident mJSN in knees without radiographically visible KOA at baseline, with an AUC of 0.75 (0.69–0.83) in the medial compartment and 0.72 (0.65–0.80) in the lateral compartment. These results were adjusted according to age, sex and BMI [33]. One limitation of this study includes its study size, which was too small for the training and validation process of its model.

In a set of 344 X-ray knee images (from 319 subjects of the well-phenotyped population of the OAI cohort), the TBT parameters were computed using fractal dimensions [37], and the fractal parameters were computed using the quadratic variations estimator (VAR) [52]. Exploiting the whole tibial proximal trabecular bone, results of this 4-year follow-up study confirmed the benefit of using TBT parameters as predictive biomarkers for the incidence of radiographic KOA with an AUC of 0.73 [0.66–0.80]. In addition, results from diagnostic odds ratios indicated that TBT-based models were able to identify a subset of true KOA initiators with a very low number of false positives [37]. The training and validation of the prediction model proposed in this study, however, was limited to using only computed radiographs. A more comprehensive study would include another acquisition modality such as the digitized X-ray films, available in the OAI cohort.

The knee joint shape using active shape modelling (NJSASM) [53] was used to assist the prediction of KOA incidence as an increase of at least 0.1 mm in mJSN or lateral JSN (lJSN) within 30 months from baseline [14]. In this study, in a set of 352 women from the PRevention of knee Osteoarthritis in Overweight Females (PROOF) study, the NJSASM-based selected descriptors, associated with other markers from clinical covariates, food questionnaires and biochemical information, were able to predict KOA mJSN and lJSN incidence with an AUC of 0.737 (0.659–0.814) and 0.731 (0.654–0.808), respectively. It should be noted that the radiological biomarkers were not among the best biomarkers for the prediction of KOA incidence in terms of an increase in KL grade, for which clinical, pain and food markers provided the best performance [14]. The use of other markers detected from radiographs would help in improving its prediction performance. 

#### 3.1.2. Increase in Kellgren–Lawrence Grade

The performance of baseline TBT-based biomarkers was also examined for predicting incident radiographic KOA in terms of Kellgren–Lawrence (KL) grade increase (∆KL ≥ 2) [14,15,34,35,36,37,38,39]. In a small set of 123 pairs of X-ray knee images from the Baltimore Longitudinal Study of Aging (BLSA) [54], the weighted neighbor distance method with a compound hierarchy of algorithms representing morphology (WND-CHARM) [39] was examined for the generation of TBT features. The WND-CHARM-based model was able to predict changes in KL grades from normal (grade 0) at baseline to minimal OA (grade 2) and moderate (grade 3) KOA at 20 years later with an accuracy of 62.4% and 72%, respectively. The most predictive bone regions were identified in locations adjacent to the tibial spines. Limitations of this study include its small study size and the lack of AUC scores, usually provided as a benchmark metric of the performance of a prediction model.

Besides TBT, other radiological biomarkers were examined for the prediction of incident radiographic KOA in terms of KL grade (KL ≥ 2), including the presence of patellofemoral joint osteoarthritis (PFJOA [34], pelvic landmarkers [35,36] and medial JSN [37]. In a set of 253 participants from the Knee Clinical Assessment Study (CAS-K) study [55], the presence of PFJOA at baseline was found to significantly increase the risk of incident radiographic tibiofemoral joint OA (TFJOA) (KL ≥ 2 and posterior osteophytes > 0 at 3 years from baseline) (OR 2.2, 95% CI 1.1 to 4.1) [34]. The use of another radiographical biomarker based on the angle between the acetabular roof and the ilium’s vertical cortex [hip α-angle] was examined for the prediction of incident radiographic KOA [36]. Examined in a set of 1184 X-ray knee images from 649 participants of the Chingford cohort [56], the performance of the prediction model based on the hip α-angle was improved (AUC = 0.797) compared to a model based only on clinical covariates (AUC = 0.692, the 95% CI values are not provided). However, the AUC 95% confidence interval values were not provided. The strength of this study includes its relatively large size cohort. It would have been interesting to evaluate the association of α-angle measurement with other parameters directly detected from knee XR images. Another radiographic biomarker based on the limb length inequality (LLI) measurement (right and left lower extremity lengths between two defined bony landmarks: the anterior superior iliac spine and the medial malleolus) was examined for the prediction of incident KOA [35] in a set of 1583 participants (2736 knees) from the Johnston County Osteoarthritis Study [57]. Results showed that the risk of KOA incidence was 20% higher among participants with LLI, but results were not statistically significant.

Finally, the VAR method was also used for the prediction of incident KOA in terms of KL increase at 48 months from baseline [37]. The TBT parameters associated with age, sex, BMI and JSN scores were found to be predictive of KOA radiographic incidence (AUC = 0.69 (0.63–0.76)) compared to using clinical covariates alone (AUC = 0.57 (0.50–0.64)).

In a group of 2628 participants from the Rotterdam cohort [58], among which 474 had KOA at endpoint, the addition of KL grade to a classical prediction model included age, sex, BMI and demographic information found to improve the performance of the prediction of KOA incidence with an AUC of 0.79 (0.77–0.81), compared to the classical model alone which achieved an AUC of only 0.67 (0.64–0.70) [15]. Tested on another set from the Rotterdam cohort or on a totally different cohort, the Chingford study [59], similar results were obtained by both KL-based and classical models [15]. Strengths of this study include its impressive study size and the validation of its model on different cohorts. It would be interesting though to perform sensitivity analysis in order to address the use of compartment-based radiographic scores such OARSI JSN scores or JSW measurements.

The ability to predict radiographic KOA incidence at 60 months, in a dataset of a relatively large set of 985 knees from the Cohort Hip and Cohort Knee (CHECK) study, was reported to be statistically significantly higher when using baseline minimum JSW and osteophyte area, in addition to demographic and clinical characteristics (AUC = 0.74 (0.69–0.78)) than without using radiographic features (AUC = 0.64 (0.59–0.68)) [38]. The performance of a model based on the use of knee alignment to predict radiographic KOA was evaluated in a set of 641 subjects from the OAI cohort. The performance of a model that used baseline knee alignment (femur–tibia angle), associated with basic clinical covariates (age, sex and BMI), previous knee injury and radiography-based parameters (KL score) in predicting KOA incidence, was found to be modest (AUC = 0.67 [0.61–0.73]) [40]. The association of knee alignment measures with KL scores did not provide a good performance, compared to using only KL scores, reported earlier for the study of Kerkhof et al. [15]. Consequently, there was no significant association between varus/valgus alignment and KOA incidence. However, when adding MRI-based WORMS and cartilage T2 scores to the abovementioned clinical and radiographic parameters, the performance was significantly better with an AUC = 0.72 [0.66–0.78].

Most of the studies considered in this section achieved a good prediction performance (AUC > 0.7). However, the assumption of independence was violated, as they did not restrict their inclusion to one knee per person. Importantly, the coefficients from a final model based on one knee per person enable independent validation in other cohorts.

### 3.2. Prediction of KOA Progression

In addition to the usual clinical covariates, such as age, gender and BMI, radiographic-based measurements, such as JSW, joint space area (JSA), OARSI JSN grade and KL grade, are commonly used for the prediction of KOA progression [27,41]. Texture analysis methods can play an important role in the improvement of the clinical-based prediction models as they provide numerical measures of KOA-induced bone changes [26]. The main specifications of the 18 studies detected for the prediction of KOA progression are summarized in Table 3. Among the 18 studies, six used TBT- based biomarkers.

#### 3.2.1. Increase in Medial Joint Space Narrowing

The performance of baseline TBT-based biomarkers was investigated for the prediction of radiographic KOA progression in terms of increase in mJSN [29,33,41,42,43,44].

In a cohort of 138 participants (248 X-ray knees) from the Prediction of Osteoarthritis Progression (POP) study, it was reported that the model based on baseline TBT parameters, extracted using fractal signal analysis (FSA), outperformed those that included baseline clinical covariates (age, sex, BMI, knee pain), bone mineral content and JSN [29] for the prediction of 36-month KOA progression, with an AUC of 0.75 [0.65–0.84] and 0.58 [0.46–0.69], respectively. This study was the first to evaluate a progression prediction model using TBT analysis of knee radiographs, although the size of radiographs included was too limited. The combination of FSA-based parameters, knee alignment and clinical covariates was more predictive of the JSN progression with an AUC of 0.79 [0.72–0.88]. In a relatively large population including 1124 patients from the Osteoarthritis Initiative (OAI) cohort [43], TBT was analyzed using VAR and Whittle, in addition to FSA. The predictive model included not only TBT parameters, but also clinical covariates (age, gender, BMI), and radiological JSN scores. Using TBT parameters, it was found that the prediction of KOA radiographic progression was improved with an AUC of 0.77 [0.73–0.80] compared to using clinical covariates alone (AUC = 0.6 [0.56–0.64]). Strengths of this study include the evaluation of three different methods of TBT analysis and the large set of radiographs included. However, the reproducibility of the regions of interest (ROIs) may not be ensured because the segmentation procedure involves the identification of the tibial spines and the lateral and medial extremities of the tibia by a trained radiologist/operator.

The VAR method was also used for the calculation of TBT features of 1888 and 683 X-ray knee images from the OAI and the Multicentre Osteoarthritis Study (MOST) cohorts, respectively [41]. The proposed prediction model included TBT features associated with CNN-based KL descriptors, as well as traditional clinical covariates and radiological JSN scores. The performance of the models was evaluated not only when training and testing on the same cohort, but also when training on one cohort (OAI or MOST) and testing on the other one (MOST or OAI). The combination of CNN-based methods and TBT analysis showed promising results in predicting radiographic progression with an AUC of 0.75 [0.71–0.79] in OAI and 0.80 [0.75–0.84] in MOST. In addition, the predictive ability of the TBT-CNN model was found to be invariant with respect to the acquisition modality or image quality. It was also shown that the prediction of KOA progression was significantly better using CNN-based KL grades (AUC = 0.67 [0.61–0.72] in OAI and AUC = 0.71 [0.66–0.76] in MOST) than those provided by radiologists (AUC = 0.63 [0.58–0.68] in OAI and AUC = 0.65 [0.60–0.71] in MOST).

Another radiographic-based method, called a dissimilarity-based multiple classifier (DMC), was used as an alternative method to calculating TBT features [44]. The DMC method uses distances between X-ray images and a diverse classifier ensemble. Based on a sample of 50 subjects, the DMC method was capable of predicting KOA 48-month radiographic progression with an accuracy of 80%, a specificity of 82% and a sensitivity of 78%. Although the size of data used in this study is small, one strength of this study includes the introduction of a fully automatic method of selecting the ROIs, based on an active-contour segmentation of tibiofemoral joint, making the TBT analysis reproducible.

The SDM method [33], previously discussed for incident JSN prediction, was also tested for the prediction of JSN progression. Using the 4-year follow-up study of Lund University, results illustrated the ability of the SDM-based model to predict medial JSN progression in knees with radiographically visible KOA (KL > 1) at baseline, with an AUC of 0.77 [0.68–0.86] in the medial compartment and 0.71 [0.63–0.79] in the lateral compartment [33]. The TBT analysis was performed on two fully-automatic selected ROIs. However, the size of each ROI was fixed and represented in pixels, which makes the TBT analysis specific to certain image resolution. It would therefore have been preferable if the ROI size had been defined according to the tibial borders detected by the active shape method used.

Another radiographic biomarker, related to the medial and lateral osteophyte scores (MLOS) in tibial plateau and femoral condyle regions, was evaluated for the prediction of KOA JSN progression in a set of 447 participants from the New York University study (204 subjects) and the OAI study (204 subjects) [42]. Although this biomarker performed poorly in predicting KOA progression, the combination of MLOS with peripheral blood leukocyte (PBL) inflammatory gene signatures showed a better performance with an AUC of 0.67 [0.59–0.74] than using MLOS alone (AUC = 0.57 [0.49–0.65]) or PBL alone (AUC = 0.62 [0.54–0.69]) [42].

#### 3.2.2. Increase in Kellgren–Lawrence (KL) Grade

The performance of baseline radiographic biomarkers was investigated for the prediction of radiographic KOA progression also in terms of increase in KL grades [34,35,46].

The CAS-K study [34] estimated the radiographic progression in tibiofemoral and patellofemoral joints. In a subgroup of 91 subjects, the results showed that the presence of mild TFJOA (posterior osteophytes = 1 or 2) in knees with KL = 2 at baseline increased the risk of PFJOA progression (KL ≥ 3 and lateral osteophytes = 3 at 3 years from baseline) (OR = 4.5 [1.8–11.2]) [34].

A radiographic biomarker based on the limb length inequality (LLI) measurement (right and left lower extremity lengths between two defined bony landmarks: the anterior superior iliac spine and the medial malleolus) was examined for the prediction of KOA progression [35] in a set of 1583 participants from the Johnston County Osteoarthritis Study [57]. The study included a group of participants with KL ≥ 2 (643 knees) at baseline. Results showed that the risk of KOA progression was significantly greater among participants with LLI > 2 cm compared to those with LLI < 2 cm (Adjusted hazard ratio of 1.83 [1.10–3.05]). This study also evaluated the risk of KOA progression in a group of participants with KL ≥ 1 (1282 knees). Knees with KL = 1 are usually considered for radiographic KOA incidence and rarely for KOA progression. Results for this group are therefore not reported in this review.

Deep convolutional neural network (CNN) learning was also used to provide radiographic biomarkers for the prediction of KL increase within the next 7 years after the baseline. The CNN-based method used a set of 2711 participants (4928 knees) from the OAI cohort as a training dataset, and 2129 subjects (3918 knees) from the MOST cohort as a testing dataset [46]. The descriptors calculated from the CNN-based method using solely knee images were found to improve the prediction performance with an AUC of 0.79 [0.87–0.81] compared to a logistic regression-based method using clinical information (age, sex, BMI, WOMAC total score, injury history, and knee injury) and radiological data (KL grades), in which the AUC was 0.75 [0.74–0.77]. The results also showed that the combination of the CNN-based descriptors with KL grades further statistically significantly improved the prediction performance with an AUC of 0.81 [0.79–0.82]. One strength of this study includes the use of a deep learning-based prediction method in which the KL scores are automatically defined directly from the raw images instead of using scores defined manually by radiologists, avoiding the subjectivity of KL grading process. This study requires high-quality graphics processing units (GPUs) to reduce the computing time during the CNN training and testing process.

Based on these studies, the benefit of the combination of CNN-based methods and classical radiographic descriptors (i.e., KL grades) was confirmed as promising biomarkers in predicting radiographic progression.

#### 3.2.3. Increase in Medial Joint Space Width

The performance of baseline TBT-based biomarkers was also investigated for the prediction of radiographic KOA progression in terms of loss in medial JSW [27,28,45].

In a group of 58 participants from the Pfizer A9001140 observational 24-month longitudinal study [60], KOA progression, defined as a loss in medial minimum JSW and medial JSA from baseline to 24 months, was evaluated using FSA of both X-ray and MRI knee images. It was found that baseline trabecular bone parameters were able to assist in the prediction of KOA progression [45]. The results showed that the proposed TBT-based model was predictive of the loss of ≥5% in medial minimum JSW and medial JSA over 24 months with an AUC of 0.85 [0.82–0.95] and 0.81 [0.79–0.85], respectively. One limitation of this study is the data size too low for the training and testing of the regression models used. In a larger population including 579 subjects (185 KOA cases with both pain and radiographic progression) from the Foundation for the National Institutes of Health (FNIH) Osteoarthritis Biomarkers Consortium, it was concluded that TBT parameters improved modestly (AUC = 0.649) but statistically significantly the prediction of KOA radiographic progression, defined as a loss of ≥0.7 mm in medial minimum JSW from baseline to 48 months, compared to using clinical covariates only (AUC = 0.608) [27]. The AUC 95% confidence interval values were not provided. Although the proposed prediction models obtained a poor AUC scores [61,62], one strength of this study is the use of well phenotyped publicly-available KOA cohort, the FNIH, that could be considered as a gold-standard cohort for comparing different prediction models.

DL-based methods have been evaluated for the prediction of the progression of radiographic JSW loss [28]. In a study including 1950 baseline X-ray images, convolutional neural networks (CNNs)-based features associated with traditional (age, sex, BMI and KL grades) risk factors were capable of predicting radiographic progression, defined as the decrease of ≥0.7 mm in medial JSW from baseline to 48 months, with an AUC of 0.857 [0.798–0.904] which was significantly higher than the traditional model (0.681 [0.608–0.748]).

Besides the studies of Kraus et al. [27,45] and Duncan et al. [34], where one knee per person was included, all other studies did not take into consideration the assumption of knee independence. Several studies have trained and validated their models using impressive number of radiographs (>1000 XRs) [28,35,41,46], whereas several other studies were based on a too limited number of radiographs, (<250 XRs) [29,33,42,44,45,63].

### 3.3. Prediction of Total Knee Replacement Risk

Total knee replacement or arthroplasty (TKA) is often considered an important clinical outcome due to the lack of disease-modifying osteoarthritis drugs (DMOADs) [63] for KOA. Besides the prediction of KOA incidence and progression, radiographic biomarkers have also been investigated for the prediction of TKA risk [47,48,49,50,51]. In this regard, two TBT-based methods were detected in this review [47,51].

The association of TBT and TKA was firstly studied using the VOT method in a limited dataset of 114 participants (28 TKA cases in 6 years after baseline). Increasing mean fractal dimension (the mean value of horizontal and vertical fractal dimension values) adjusted for age, sex, BMI, JSN grade and WOMAC score was found to reduce the odds of TKA [47], independent of radiographic KOA disease. The medial tibiofemoral osteophyte score was also found to be a significant predictor of TKA (OR = 2.0, 95% CI [1.27, 3.13], *p* = 0.003). A limitation of this study is the small sample size.

Very recently, the ability of radiographic TBT features measured using the VAR method was evaluated, for the first time, for the prediction of TKA risk [51]. Associated with radiographic severity descriptors in a set of 4382 participants (375 cases) from the OAI cohort, the TBT-based prediction model achieved an AUC of 0.92 (0.90–0.93) while a reference model based on standard clinical covariates and radiological KL grades achieved an AUC of 0.86 (0.84–0.86). The results also showed that the TBT-based model was able to identify at-risk patients with a 60% increase in TKA case prediction compared to the reference model, as reflected by the recall values.

The use of 2-year changes in medial JSW at minimum (mJSW) and fixed (fJSW) knee joint positions has also been evaluated for the prediction of TKA risk within a subsequent 7-year follow-up period [48], in a set of 627 participants (107 cases) from the OAI cohort. The performance of the prediction models based on JSW measures (AUC = 0.57 [0.50–0.64] for mJSW and 0.61 [0.55–0.68] for fJSW) was similar to that based on quantitative MRI femorotibial cartilage thickness (AUC = 0.62 [0.55–0.68]). The performance of this study, represented by AUC, is the lowest among the published studies in TKA risk prediction, as seen in Table 4.

DL-based methods have also been evaluated for the prediction of TKA risk [50], in a set of 728 participants from the OAI cohort. In this study, one-to-one case-control matching, based on clinical variables of age, sex, BMI and ethnicity, was used and 364 cases were consequently included. The prediction model based on DL achieved an AUC of 0.87 (0.85–0.90), outperforming a reference model based on KL grades alone which achieved an AUC of 0.74 [0.71–0.77].

The use of KL grade, combined with clinical covariates (age, sex, BMI, urinary cross-linked C-terminal telopeptide of type II collagen (uCTX-II)), was evaluated for the prediction of TKA risk at 24 months [49]. In a set of 935 knees, the performance of the combined model was statistically significantly improved (AUC = 0.75 [0.72–0.77]), compared to the model without KL grade (AUC = 0.69 [0.67–0.72]). Due to the short period of followup (24 months), the incidence of TKA was very low (2%).

## 4. Discussion

This review shows that the three most important endpoints (incidence, progression and TKA) related to KOA have already been widely investigated and validated using radiographic biomarkers. These are summarized and discussed in terms of KOA endpoints and are further discussed in connection with JSN or KL changes several years from baseline. Results showed that prediction models are more effective in progression than in incidence in terms of AUC scores. This is understandable because when the disease exists, it is easier to predict its progression than when there are only prodromes of the disease, as observed in Parkinson’s [64] and stroke [65] diseases.

Radiographic biomarkers were used not only in prognostic multivariable modeling studies, where their longitudinal relationship (prediction) with an outcome is searched, but also in diagnostic multivariable modeling studies [25,66,67,68], where a cross-sectional relationship (detection) with an outcome is searched for. However, in this review, the focus was only on studies involving the longitudinal relationship between radiographic biomarkers and either KOA incidence, progression or TKA incidence.

KOA incidence is usually defined as knees with no radiographic KOA at baseline and moderate or severe KOA at followup, whereas KOA progression is defined as knees with moderate radiographic KOA at baseline and higher radiographic scoring at followup [69,70].

For the prediction of KOA incidence, the majority of the studies reviewed included knees with KL ≤ 1 at baseline, except for a few studies where knees with no radiographic (KL = 0) KOA were included [14,37,39]. However, for the prediction of KOA progression, although knees with moderate KOA at baseline were intentionally targeted, there was no specific inclusion criterion that was mostly employed. As reported in Table 3, the criterion for knee inclusion was knees with KL = 1, 2 or 3 in three studies [27,29], with KL = 2 or 3 in five studies [28,41,42,43,45], or with KL ≥ 1 in three studies [44,46], and with KL ≥ 2 in two studies [33,35]. In order to have fair and valid comparisons of the performance of the proposed KOA prediction models, the KOA community should identify a unique definition of both KOA incidence and progression as well as standardize a specific inclusion criterion.

The (AUC) has been recommended and preferred for overall accuracy for the evaluation of machine learning algorithms [71]. Consequently, in Table 2, Table 3 and Table 4 the AUC values were provided in order to inform readers as completely as possible of this predictive performance criterion. However, it should be pointed out that it was not possible to strictly compare these values from one study to another because the databases studied differ; the inclusion criteria and the definition of the endpoints are also different.

Theoretically, an AUC of 0.5 means no discrimination (i.e., inability to diagnose patients with and without the disease or condition based on the test), 0.7 to 0.8 is considered acceptable, 0.8 to 0.9 is considered excellent, and more than 0.9 is considered outstanding [61,62].

Among the studies reviewed, 5 out of 11 provided acceptable AUC scores (≥0.7) for the prediction of radiographic KOA incidence and 7 out of 12 for the prediction of radiographic KOA progression. Three out of five studies provided acceptable AUC scores for the prediction of TKA risk. Theoretically, the prediction performance related to TKA should be lower than that related to KOA progression because there are more parameters to be considered to simulate the complexity of TKA as an outcome of KOA, including, for example the opinion of doctors and patients in the final decision of TKA [51]. Surprisingly, the overall AUC scores were better for the prediction of TKA than KOA progression. However, including knees with KL = 4 was found to be highly associated with TKA risk [50,51].

The regions of interest in the knee tibial or femoral compartments also play an important role in the performance of the KOA prediction models, especially when using TBT analysis. For instance, the medial subchondral tibial region was found to be the most predictive of KOA incidence [33] and progression [27,29]. Conversely, the regions adjacent to the tibial spines were found to be predictive of KOA progression [39]. In addition, and as demonstrated previously for the prediction of both KOA progression and TKA risk, the most predictive TBT features were provided not only by the medial subchondral bone region, but also by lateral regions [43,51]. In later studies, the most predictive TBT features were found in both subchondral cortical bone and subchondral trabecular bone located in distal regions. This observation might be explained by the complex interactions and cross-talk between cartilage and bone tissue, but also by the role of bone biomechanics properties in the model of knee OA progression [37].

A greater baseline WOMAC pain score was found to be associated with KOA incidence and progression in a diagnostic multivariable modeling study [72]. However, in a prognostic multivariable modeling study, knee pain or WOMAC pain showed only a moderate performance for predicting KOA progression [73]. Furthermore, the performance of a model including TBT parameters, age, sex, BMI, KL and JSNM to predict KOA progression remained unchanged when adding WOMAC pain [41].

As shown in Table 2, Table 3 and Table 4, the capability of radiographic biomarkers to predict changes in KL or JSN grade has been evaluated in different well-known KOA cohorts (OAI, MOST, FNIH, CHECK, …) and found acceptable. However, researchers have recently been interested in evaluating the capability of radiographic biomarkers to predict changes in articular cartilage degeneration assessed by MRI [74,75]. For example, the relation between Xray-based TBT features and cartilage composition assessed by MRI was evaluated [74] and a weak relationship between radiographic TBT features extracted from the medial subchondral bone and MRI-based T2 relaxation time values of the medial tibial cartilage was found [74]. In addition, the combination of baseline TBT parameters and 18-month variations in MRI subchondral bone texture score was found to be significantly associated with radiographic progression at 36 months [76]. Furthermore, the 3D MRI bone texture was recently evaluated for its association to TKA [77]. The MRI-based femoral bone shape was also found to help predict incident TKA [78] or KOA progression [79]. The combination of radiographic biomarkers and MRI-based biomarkers, such as cartilage WORKS score, were also evaluated for the prediction of radiographic KOA progression [40,42,80]. However, radiographic biomarkers may have a stronger role in TKA risk screening compared to MRI-based biomarkers, considering the lower cost and easier implementation of radiography in primary care practice [26].

Several studies evaluated their proposed prediction models on a relatively limited sample size (<150 subjects) cohort [29,33,34,39,45,47]. It would therefore be interesting to validate their results on a larger set of KOA cohorts.

A new challenge in the field of KOA management has been identified which consists in defining phenotypes of OA such as subchondral bone, metabolic syndrome, synovitis, mechanical injury, aging and cartilage driven KOA [81]. Thus, it should be relevant to evaluate the capacity of imaging prediction models in assessing the risk of progression of KOA for specific identified phenotypes.

Except for two studies, all the other studies reviewed proposed prediction models that were trained and tested in the same cohort. One study proposed a model based on the use of KL grades for the prediction of radiographic KOA incidence [15]. This model was first trained in a set from the Rotterdam cohort and then tested in another set of the Rotterdam cohort and in a totally different cohort, the Chingford cohort. The other study proposed a model based mainly on the use of TBT features for the prediction of radiographic KOA progression [41]. This model was not only trained and tested in the same cohort, but also trained in one cohort and tested on a totally different cohort. On one part, the TBT-based model was trained in the OAI cohort and tested in the MOST cohort, and on the other part, the TBT-based model was trained in the MOST cohort and tested in the OAI cohort.

As a perspective for future work by the KOA research community, it would be of great interest to define a longitudinal dataset, from several cohorts (e.g., OAI, MOST, CHECK, PROOF, etc.). This would make it possible to examine and fairly compare the impact of the different imaging biomarkers proposed, separately or in a composite way, on the performance of the models for predicting radiographic KOA incidence and progression, as well as the TKA risk. The KOA research community is also encouraged to work on the identification of a unique definition of both radiographic KOA incidence and progression.

## Figures and Tables

**Figure 1 life-13-00237-f001:**
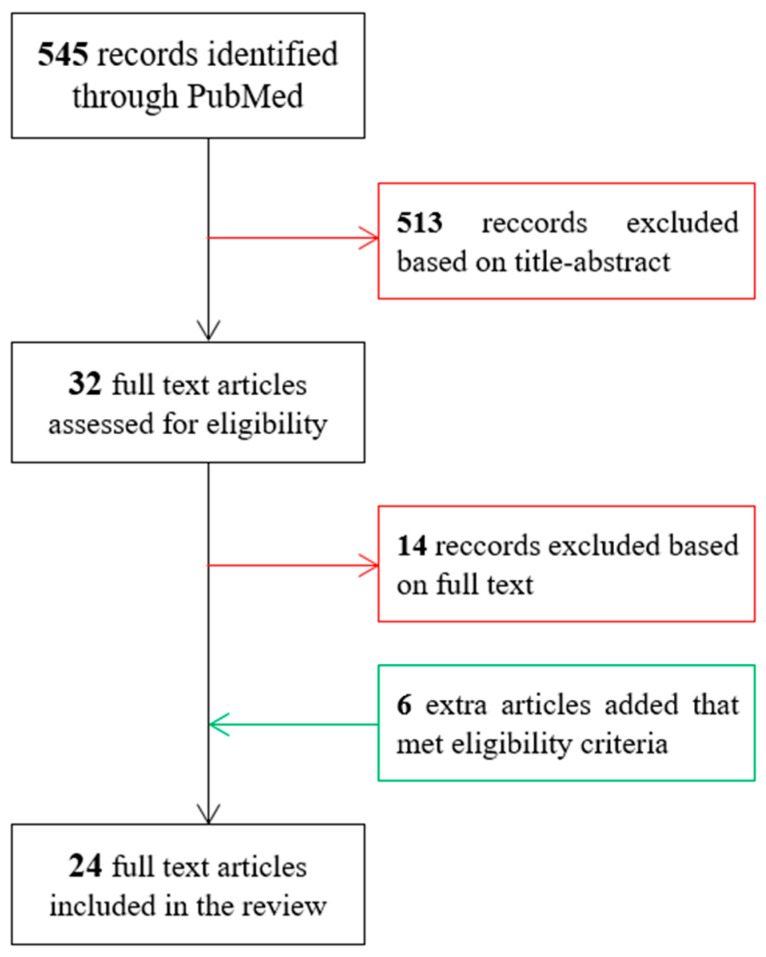
Flow diagram of the article selection.

**Table 1 life-13-00237-t001:** List of studies included in the three categories considered.

Objective	Included Studies	Nubmer of Studies
Prediction of KOA incidence	[14,15,33,34,35,36,37,38,39,40]	10
Prediction of KOA ^1^ progression	[27,28,29,33,34,35,41,42,43,44,45,46]	12
Prediction of TKA ^2^ risk	[47,48,49,50,51]	5

^1^ Knee OsteoArthritis. ^2^ Total Knee Arthroplasty.

**Table 2 life-13-00237-t002:** Summary of the major studies related to the prediction of knee osteoarthritis radiographic incidence.

Authors (Publication Year, Reference)	Cohort Name (number of Subjects, % of Cases)	Period (Months)	Inclusion Criterion	Incidence Definition	Main Radiographic Biomarkers	Best AUC
Garriga et al. (2020) [36]	Chingford (649)	48	KL ^8^ ≤ 1, JSN < 1	KL ≥ 2	Hip α-angle	0.80
Joseph et al. (2018) [40]	OAI ^1^ (641, 13%)	72	KL ≤ 2	KL > 2	KL & KA ^13^	0.67
Janvier et al. (2017) [37]	OAI (319, 13%)	48	KL = 0	ΔJSN ^10^ ≥ 1	TBT ^14^	0.73
Janvier et al. (2017) [37]	OAI (319, 13%)	48	KL = 0	ΔKL ≥ 1	TBT	0.69
Lazzarini et al. (2017) [14]	PROOF ^2^ (352,11%)	30	KL = 0	mJSN ^11^ ≥ 1 mm	NJSASM ^15^	0.74
Lazzarini et al. (2017) [14]	PROOF (352,12%)	30	KL = 0	lJSN ^12^ ≥ 1 mm	NJSASM	0.73
Kerkhof et al. (2014) [15]	Rotterdam (2628,18%)		KL ≤ 1	KL ≥ 2	KL	0.79
Kinds et al. (2012) [38]	CHECK ^3^ (653, 20%)	60	KL ≤ 1	KL ≥ 2	JSW ^16^ & OSTA ^17^	0.69
Woloszynski et al. (2012) [33]	LU ^4^ (105, 34%)	48	KL ≤ 1	ΔJSN ≥ 1	TBT	0.75
Duncan et al. (2011) [34]	CAS-K ^5^ (253, 22%)	36	KL = 0 or 1, PO ^9^ = 0	KL ≥ 2 or PO > 0	TFJOA ^18^	-
Golightly et al. (2010) [35]	JCO ^6^ (2734 *, 15%)	36–156	KL ≤ 1	KL ≥ 2	LLI ^19^	-
Shamir et al. (2009) [39]	BLSA ^7^ (123, 32%)	240	KL = 0	KL ≥ 2	TBT	-

Acronyms refer to the following: ^1^ OAI—Osteoarthritis Initiative Study. ^2^ PROOF—The PRevention of knee Osteoarthritis in Overweight Females. ^3^ CHECK—Cohort Hip and Cohort Knee study. ^4^ LU—Lund University study. ^5^ CAS-K—Knee Clinical Assessment study. ^6^ JCO—Johnston County Osteoarthritis Study. ^7^ BLSA—Baltimore Longitudinal Study of Aging. ^8^ KL—Kellgren–Lawrence grade. ^9^ PO—posterior osteophytes. ^10^ JSN—OARSI medial or lateral joint space narrowing score. ^11^ mJSN—OARSI medial joint space narrowing score. ^12^ lJSN—OARSI lateral joint space narrowing score. ^13^ KA—Knee alignment. ^14^ TBT—trabecular bone texture. ^15^ NJSASM—knee joint shape using active shape modelling. ^16^ JSW—minimum joint space width. ^17^ OSTA—osteophyte area. ^18^ TFJOA—tibiofemoral joint osteoarthritis. ^19^ LLI—limb length inequality. * Number of knees.

**Table 3 life-13-00237-t003:** Summary of the major studies related to the prediction of knee osteoarthritis radiographic progression.

Authors (Publication Year, Reference)	Cohort name (number of Subjects, % of cases)	Period (Months)	Inclusion Criterion	Progression Definition	Main Radiographic Biomarkers	AUC
Almhdie-Imjabbar et al. (2022) [41]	OAI ^1^ (1888, 16%)	48	1 < KL ^10^ < 4	ΔmJSN ^14^ ≥ 0.5	TBT ^16^, JSN	0.75
Almhdie-Imjabbar et al. (2022) [41]	MOST ^2^ (683, 36%)	60	1 < KL < 4	ΔmJSN ≥ 0.5	TBT, JSN	0.80
Guan et al. (2020) [28]	OAI (1950, 48%)	48	1 < KL < 4	ΔJSW ≥ 0.7	KL, CNNf ^17^	0.86
Attur et al. (2020) [42]	OAI (204, 30%) NYU ^3^ (243, 30%)	24	1 < KL < 4	ΔmJSN ≥ 0.5 mm	MLOS ^18^	0.67
Tiulpin et al. (2019) [46]	OAI (2711, 27%) MOST (3918, 47%)	60	KL ≥ 1	ΔKL ≥ 1	KL, CNNf	0.81
Kraus et al. (2018) [27]	FNIH ^4^ (579, 32%)	24–48	0 < KL < 4, JSN ^11^ < 2	ΔJSW ≥ 0.7	TBT	0.65
Janvier et al. (2017) [43]	OAI (1124, 14%)	48	1 < KL < 4	ΔmJSN ≥ 1	TBT, JSN	0.77
Kraus et al. (2013) [45]	Pfizer (58, 36%)	12–24	1 < KL < 4, JSW ^12^ ≥ 2 mm	ΔJSW ≥ 5% ΔJSA ^15^ ≥ 5%	TBT	0.85
Woloszynski et al. (2012) [44]	UWA ^5^ (50, 24%)	48	KL > 1	ΔmJSN ≥ 1	TBT	-
Woloszynski et al. (2012) [33]	LU ^6^ (105, 27%)	48	KL ≥ 2	ΔJSN ≥ 1	TBT	0.77
Duncan et al. (2011) [34]	CAS-K ^7^ (91, 25%)	36	KL = 2, PO ^13^ =1 or 2	KL ≥ 3 or PO = 3	PFJOA ^19^	-
Golightly et al. (2010) [35]	JCO ^8^ (1282 *, 34%)	36–156	KL ≥ 1	ΔKL ≥ 1	LLI ^20^	-
Golightly et al. (2010) [35]	JCO (643 *, 27%)	36–156	KL ≥ 2	ΔKL ≥ 1	LLI	-
Kraus et al. (2009) [29]	POP ^9^ (138, 13%)	36	0 < KL < 4	ΔmJSN ≥ 1	TBT, KA ^21^	0.79

Acronyms refer to the following: ^1^ OAI—Osteoarthritis Initiative Study. ^2^ MOST—Multicentre Osteoarthritis Study. ^3^ NYU—New York University study. ^4^ FNIH—Foundation for the National Institutes of Health study. ^5^ UWA—University of Western Australia study. ^6^ LU—Lund University study. ^7^ CAS-K—Johnston County Osteoarthritis Study. ^8^ JCO—Johnston County Osteoarthritis Study. ^9^ POP—Prediction of Osteoarthritis Progression Study. ^10^ KL—Kellgren–Lawrence grades. ^11^ JSN—joint space narrowing. ^12^ JSW—joint space width. ^13^ PO—posterior osteophytes. ^14^ JSN—medial or lateral joint space narrowing. ^15^ JSA—joint space area. ^16^ TBT—trabecular bone texture. ^17^ CNNf—Radiographic features detected by conventional neural networks. ^18^ MLOS—medial and lateral osteophyte scores in tibial plateau and femoral condyle regions. ^19^ PFJOA—patellofemoral joint osteoarthritis. ^20^ LLI—limb length inequality. ^21^ KA—Knee alignment. * Number of knees.

**Table 4 life-13-00237-t004:** Summary of the main studies related to the prediction of total knee replacement risk.

Authors (Publication Year, Reference)	Cohort Name (Number of Subjects, % of Cases)	Period (Months)	Inclusion Criterion	Main Radiographic Biomarkers	AUC
Almhdie-Imjabbar et al. (2022) [51]	OAI ^1^ (4382, 9%),	108	0 ≤ KL ^4^ ≤ 4,	TBT ^7^ & KL & JSN ^8^	0.92
Almhdie-Imjabbar et al. (2022) [51]	OAI (4296, 7%)	108	0 ≤ KL ≤ 3	TBT ^7^ & KL & JSN	0.86
Leung et al. (2020) [50]	OAI (728, 50%)	108	0 ≤ KL ≤ 4	KL, RNetF ^9^	0.87
Kwoh et al. (2020) [48]	OAI (627, 17%)	82	2 ≤ KL ≤ 3	JSW	0.61
Bihlet et al. (2020) [49]	NCT ^2^ (935, 2%)	24	2 ≤ KL ≤ 3 JSW ^5^ ≥ 2.0 mm	KL	0.75
Podsiadlo et al. (2014) [47]	ACHMA ^3^ (114, 25%)	72	0 ≤ KL ≤ 3 OST ^6^ ≥ 1	TBT	-

Acronyms refer to the following: ^1^ OAI—Osteoarthritis Initiative Study. ^2^ Nordic Bioscience A/S two randomized, double-blind, multi-center, placebo-controlled phase III trials (NCT00486434 & NCT00704847). ^3^ ACHMA—Alfred and Caulfield Hospitals in Melbourne, Australia. ^4^ KL—Kellgren–Lawrence grades. ^6^ OST—OARSI osteophyte score. ^5^ JSW—joint space width. ^7^ TBT—trabecular bone texture. ^8^ JSN—joint space narrowing. ^9^ RNetF—Radiographic features detected by Residual neural network (an artificial neural network).

## Data Availability

Data is contained within the article and Appendix A.

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
