# Peer review of "Radiographic Biomarkers for Knee Osteoarthritis: A Narrative Review"

_life, 2023, doi:10.3390/life13010237_

Round 1

Reviewer 1 Report

·         The related works section is very short, and no benefits from it. I suggest increasing the number of studies and adding a new discussion to show the studied works' advantages, disadvantages, and weaknesses. The authors should discuss the literature review more deeply and clearly.

·         Please use the complete form of abbreviations in the abstract.

·         The abstract also should be revised according to the main idea of this research and the main motivation behind the proposed research.

·         The research findings and contribution need to be stated clearly. As well as the obtained review in this paper.

·         The authors need to use more passive voice

·         Redundant use of the word "our" throughout the paper

·         Miss use of definite and indefinite articles

·         The paper has to be proofread by a native English reviewer.

·         Authors should also include more scientific logic in explaining the review results are previous work. 

Reviewer 2 Report

General comments

The article Radiographic biomarkers for Knee Osteoarthritis: a narrative review is well written and well organized. In this review, the authors summarized the current state of knowledge regarding human studies of radiographic imaging biomarkers used to predict outcomes associated with knee osteoarthritis (KOA). The methodology of the adopted method was described in a concise and clear manner. The authors present a review paper, and the literature list contains 71 items. In my opinion, this is far too few for this type of article. Conclusions are presented in a concise manner.  In my opinion, it seems necessary to make minor corrections to allow a more thorough understanding of the topic. The following are my comments.

Minor comments:

Introduction

The incidence of osteoarthritis is influenced by many factors, such as work, sports participation, musculoskeletal injuries, obesity and gender. Information about this, along with the necessary literature, should be added after second paragraph of the introduction. Authors may find some useful information in the works: DOI 10.1016/S0140-6736(19)30417-9; DOI10.3390/app11041552; https://doi.org/10.1136/annrheumdis-2013-204763; doi:10.1038/nrrheum.2015.135 DOI 10.3390/app10238312; https://doi.org/10.4081/or.2014.5188; doi:10.1038/nrrheum.2015.135;DOI: 10.1056/NEJMcp1903768;

The introduction should be expanded to include more information on typical diagnostic methods (CT, MRI, ultrasound) including physical examination, as well as alternative methods such as vibroarthrography with limitations in the diagnosis of osteoarthritis. Authors may find some useful information in the works:

https://doi.org/10.1016/j.cpet.2018.08.004; https://doi.org/10.1111/j.1617-0830.2006.00063.x; DOI 10.3390/app9194102; https://doi.org/10.1016/j.berh.2016.09.007; doi:10.35784/acs-2022-14; https://doi.org/10.3390/s22062176; https://doi.org/10.3390/s22103765;

Please make the relevant additions with the necessary literature. This will allow you to better understand the topic and highlight the essence of the issue at hand.

Results and Discussion

The chapters on the results and the discussion are presented correctly. The compilation of the results in tables greatly facilitates the reading of the work. In my opinion, these chapters do not need improvement and can be left in their current form.

After making the appropriate additions, the article may be accepted for publication.

Round 2

Reviewer 1 Report

·         Redundant use of the word "our" throughout the paper

·         Miss use of definite and indefinite articles

·         The paper has to be proofread by a native English reviewer.

·         Authors should also include more scientific logic in explaining the experimental results.

·         In general, update the references lists by the following reference related to:

·         After that, you should mention that these methods can be used to optimize the problem.

·         What other possible methodologies can be used to achieve your objective in relation to this work?

Cite this paper

Sampath Dakshina Murthy, Achanta, Thangavel Karthikeyan, and R. Vinoth Kanna. "Gait-based person fall prediction using deep learning approach." Soft Computing (2021): 1-9. https://doi.org/10.1007/s00500-021-06125-1.
